# Impact of sampling depth on pathogen detection in pit latrines

**Drew Capone** [1,2], **Petros Chigwechokha** [3], **Francis L. de los Reyes, III** [4], **Rochelle H. Holm** [5,6], **Benjamin B. Risk** [7], **Elizabeth Tilley** [8,9], **Joe Brown** [1,2] *

**1** Civil and Environmental Engineering, Georgia Institute of Technology, Atlanta, Georgia, United States of America, **2** Department of Environmental Sciences and Engineering, Gillings School of Public Health, University of North Carolina at Chapel Hill, Chapel Hill, North Carolina, United States of America, **3** Directorate of Research, Malawi University of Science and Technology, Blantyre, Malawi, **4** Department of Civil, Construction, and Environmental Engineering, North Carolina State University, Raleigh, North Carolina, United States of America, **5** Centre of Excellence in Water and Sanitation, Mzuzu University, Mzuzu, Malawi, **6** Christina Lee Brown Envirome Institute, University of Louisville, Louisville, Kentucky, United States of America, **7** Department of Biostatistics and Bioinformatics, Emory University, Atlanta, Georgia, United States of America, **8** Department of Environmental Health, University of Malawi, Blantyre, Malawi, **9** Department of Sanitation, Water and Solid Waste for Development, Swiss Federal Institute of Aquatic Science and Technology, Duebendorf, Switzerland

* joebrown@unc.edu

**Data Availability Statement:** All data used in this paper is available at a dedicated data repository at Open Science Framework (OSF.io). Permanent link: https://osf.io/5gn9u/.

## Abstract

Wastewater based epidemiology (WBE) is increasingly used to provide decision makers with actionable data about community health. WBE efforts to date have primarily focused on sewer-transported wastewater in high-income countries, but at least 1.8 billion people in low- and middle-income countries (LMIC) use onsite sanitation systems such as pit latrines and septic tanks. Like wastewater, fecal sludges from such systems offer similar advantages in community pathogen monitoring and other epidemiological applications. To evaluate the distribution of enteric pathogens inside pit latrines–which could inform sampling methods for WBE in LMIC settings unserved by sewers–we collected fecal sludges from the surface, mid-point, and maximum-depth of 33 pit latrines in urban and peri-urban Malawi and analyzed the 99 samples for 20 common enteric pathogens via multiplex quantitative reverse transcription PCR. Using logistic regression adjusted for household population, latrine sharing, the presence of a concrete floor or slab, water source, and anal cleansing materials, we found no significant difference in the odds of detecting the 20 pathogens from the mid-point (adjusted odds ratio, aOR = 1.1; 95% confidence interval = 0.73, 1.6) and surface samples (aOR = 0.80, 95% CI = 0.54, 1.2) compared with those samples taken from the maximum depth. Our results suggest that, for the purposes of routine pathogen monitoring, pit latrine sampling depth does not strongly influence the odds of detecting enteric pathogens by molecular methods. A single sample from the pit latrines' surface, or a composite of surface samples, may be preferred as the most recent material contributed to the pit and may be easiest to collect.

**Funding:** This study was funded in part by the Bill and Melinda Gates Foundation (www. gatesfoundation.org) grant OPP1137224 to JB. The funders had no role in study design, data collection and analysis, decision to publish, or preparation of the manuscript.

**Competing interests:** The authors have declared that no competing interests exist.

## Author summary

Almost two billion people defecate into onsite sanitation facilities such as pit latrines. Evidence suggests these individuals are often at a disproportionately high risk of enteric infection compared to those served by piped sewerage. Public health surveillance in high-risk communities could monitor key pathogens in fecal sludges to inform which pathogens are circulating among those contributing waste. Representative surveys including multiple latrines could inform vaccination campaigns and interventions to limit the transmission of these pathogens, including eradication efforts, at a localized scale. A better understanding of how enteric pathogen detection may vary by pit latrine sampling depth would be useful to inform methods for fecal sludge collection. To refine methods for sampling in such surveys, we collected fecal sludges from the surface, mid-point, and maximum depth of pit latrines in Malawi. We found pit latrine depth did not significantly affect enteric pathogen detection by the methods we used. However, surface samples may be preferred due to their ease of collection and because the most recently deposited fecal material may be of greater interest in surveillance efforts that aim to inform pathogen carriage at the time of sampling.

## Introduction

Wastewater-based epidemiology (WBE) is a rapid and scalable approach with the potential to conduct community health surveillance in near real-time [1–4]. As a complement to other public health surveillance efforts, WBE has provided actionable data to decision makers in global polio eradication efforts [5–8] and in response to the COVID-19 pandemic [2,9–12]. Other WBE studies, primarily focused on wastewater from high-income countries, have investigated illicit and licit drug use [13,14], antibiotic resistance [15], the community microbiome [16], and a range of biomarkers that may be useful in understanding other aspects of community health [17,18].

In low- and middle-income countries (LMIC), onsite sanitation technologies–including pit latrines and septic tanks–serve at least 1.8 billion people [19,20]. In most cities in the Global South, sewers serve a small proportion of the population and often do not extend to unplanned urban settlements where disease burdens may be disproportionately high [21,22]. Such settings may also lack capacity for broad clinical surveillance of infections [23], hindering the development and application of pathogen-specific intervention strategies. Populations that are unserved by sewerage systems may therefore be among those that could benefit most from systematic waste monitoring to inform public health measures to reduce risk.

Sampling fecal sludges from onsite sanitation technologies in low-income settings is a plausible and logical extension of current approaches to WBE using wastewater from centralized systems [24]: sludges represent composite waste from a group of individuals, though typically fewer than wastewater [25,26]. Because fecal sludges can be sampled from shared latrines, pit emptying services (tanker trucks), disposal sites representing wastes from many latrines, or using representative sampling from multiple household onsite facilities, fecal sludges can offer a pooled, noninvasive, potentially anonymized community sampling approach that is logistically far easier than collection of biological specimens from a representative number of individuals. Apart from sampling strategies to inform community surveillance, there are also applications where the provision of data at more limited scales (e.g., clusters of households, small neighborhoods, or other defined areas of special interest) may be preferred over a snapshot of an entire sewershed (i.e., the area served by a sewer network). A more inclusive term

for these approaches may be "waste-based epidemiology", retaining the WBE acronym but acknowledging that fecal waste sampling can be much broader than wastewater from sewerage systems only. Such approaches may be feasible in cities that remain largely unserved by sewerage [27] where risks may be highest [28,29], complementing other health measurement strategies.

Ideally, a representative sample of fecal sludge from an onsite system would be a composite of the feces from all individuals contributing waste to the system and would be free of solid waste and other non-target material. This is challenging, however, because sludges in pit latrines may not be well mixed and often contain refuse and anal cleansing materials [30]. The extent of mixing of materials within onsite containment systems has not been previously characterized, but likely depends on multiple factors including latrine type, operation, emptying frequency, and the level of the water table [26]. For example, pour-flush latrines, or latrines that receive greywater as an input, may experience greater mixing compared to dry pit latrines which function without water.

Because onsite sanitation facilities may be highly diverse within and across settings of interest, extending the principles of WBE to fecal sludges requires standardized methods for sampling. The physical [31,32] and microbial [32–34] dynamics of fecal sludges have increasingly been studied, but attempts to characterize the spatial distribution of enteric pathogens in onsite technologies have been primarily limited to fecal indicator bacteria [35]. It is unclear how the intra-pit depth (e.g. surface, mid-point, maximum depth) of fecal sludge sampling may impact the detection of enteric pathogens in high burden settings, a key application of the approach. To explore this gap, the aim of this study was to compare molecular detection of enteric pathogens between the surface, mid-point, and maximum depth of dry pit latrines in urban and peri-urban Malawi. Understanding how enteric pathogen detection varies with pit depth may inform sampling methods for pathogen surveillance in LMICs, which could provide actionable data to guide prioritization of public health interventions such as vaccination [36], mass drug administration [37], or improved water, sanitation, and hygiene (WASH) [38].

## Methods

### Ethics statement

Participation was based on informed, written consent by a household member over the age 18. This study received ethical approval and a material transfer agreement from the Republic of Malawi National Health Sciences Research Committee (Protocol 17/09/1915).

We purposively enrolled participants in urban Blantyre and peri-urban Mzuzu, Malawi based on the condition of their onsite sanitation technology. The following criteria were used for enrollment: (A) the presence of a pit latrine with a concrete or wood plank floor or slab, (B) a lined pit below ground, (C) drop hole large enough for a Gulper (a type of manual pump used to empty pit latrines) [26] to empty, (D) sufficiently watery sludge capable of being pumped by a Gulper, and (E) less than 1 meter of space between the surface of the fecal sludge and the latrine floor.

### Sample collection

We collected fecal sludge from the surface, mid-point, and maximum depth of each pit in the urban Ndirande neighborhood in Blantyre and the peri-urban areas surrounding Mzuzu, Malawi from November to December 2018, which is the beginning of the rainy season. First, a sludge sample was scooped from the surface of each pit using a disposable hand scoop placed through the squat hole (S1 Fig). If there was visible solid waste in the pit that was likely to clog

the Gulper, the waste was carefully removed using a metal pitchfork while attempting to minimize any mixing of the pit contents (S2 Fig). Next, a Gulper was lowered into the pit until the full length of the Gulper (approximately 1.5m) was submerged (S3 Fig). The first "pump" of sludge was assumed to have been the sludge that filled the Gulper as it was lowered and was not saved. Then, the second "pump" of sludge was collected in a bucket and considered as the sludge from the maximum depth of the pit. When approximately half of the pit was emptied, another "pump" of sludge was collected into a separate bucket and considered as the sludge to be from the mid-point of the pit. Finally, the sample from each depth was aliquoted into three sterile 15 milliliter (mL) centrifuge tubes which were transported to the lab at Mzuzu University (Mzuzu samples) or the University of Malawi (Blantyre samples) (S4 Fig).

### Data collection

On the same day as sample collection from the household's latrine, we interviewed an adult member of the household regarding the characteristics and typical use of their pit latrine. Direct observation of the characteristics and condition of the latrine was guided by a checklist.

### Sample processing

We stored samples temporarily at 4˚ Celsius (C) until all samples had been collected (about ten days). Then, the 15mL tubes were centrifuged at 2000 revolutions per minute (RPM) for 5 minutes, the supernatant was poured off, 0.5 gram (g) of each pellet was transferred into a sterile 2mL centrifuge tube, and then 0.75mL of UNEX buffer (Microbiologics, St. Cloud, Minnesota, USA) was added to each tube. Samples were shipped at ambient conditions to North Carolina State University at Raleigh (Raleigh, North Carolina) and then transported on ice to Georgia Institute of Technology (Atlanta, Georgia), where they were transferred to a -80˚C freezer for storage until extraction.

We extracted total nucleic acids from 100 milligrams (mg) of the fecal sludge-UNEX buffer mixture using the Qiagen 96 Virus QIAcube HT Kit (Qiagen, Hilden, Germany) (S1 Text) [24,39]. Samples were spiked with bacteriophage MS2 (a single-stranded RNA virus) as an extraction control and one negative extraction control was included per day of extractions. Then, nucleic acids were analyzed with a custom TaqMan Array Card (TAC)–a multiplex quantitative reverse transcription PCR platform–that tested for 20 enteric pathogens in duplicate [24] (S2 Text, S1 and S2 Tables). Each TAC included a positive and negative control; positive controls were gene targets inserted into plasmids [40]. Exponential curves and multicomponent plots were examined for each target–including comparison with the positive control–to validate positive amplification. Any sample exhibiting positive amplification in either duplicate well before a quantification cycle (Cq) of 40 was called positive [24].

### Data analysis

We compared individual pathogen detection between depths using the Jaccard similarity coefficient [41] (i.e., the intersection of detections over the union). Because it excludes instances where a pathogen is not detected in either depth, this coefficient does not bias the estimate upwards for infrequently detected pathogens. For example, from the matched 33 surface and 33 mid-point samples, if we were to detect an individual pathogen at both depths 20 times, only in the surface 4 times, only in the mid-point 6 times, and in neither sample 3 times, the Jaccard similarity coefficient is 67% (Eq 1). Further, we calculated 95% confidence intervals around the Jaccard similarity coefficient by bootstrapping our data with 10,000 iterations.

$$J_X = 20 \div (20 + 4 + 6) = 67\% \tag{1}$$

We used logistic regression to compare overall pathogen detection between pit latrine depths which generated unadjusted and adjusted prevalence odds ratios (OR, aOR). The binary presence or absence of each of the 20 pathogens assessed was our response variable, such that each observation represented the presence or absence of a particular pathogen at a given depth and latrine; depth as a categorical variable (maximum depth, mid-point, surface) was our exposure variable. We *a priori* selected potential confounders from our household survey questionnaire based on the potential to impact pathogen detection in fecal sludges, estimating both unadjusted and adjusted effects. Selected confounders included household population, whether the pit latrine was shared with other households, the presence of a concrete floor or slab, a categorical variable for water source (public tap, tap inside the compound, tap inside the house), and the most frequent anal cleansing material used (water or paper). We included random effects for pit latrine (33 levels), pathogen (20 levels), and their interaction. To fit regression models we used generalized linear mixed-effects models (GLMM) with a binomial distribution (logit link) [42]. We analyzed data in R version 4.0.0 (R Foundation for Statistical Computing, Vienna, Austria).

## Results

We collected fecal sludge from the surface, mid-point, and maximum depth of 3 dry pit latrines in peri-urban Blantyre and from 30 dry pit latrines in peri-urban Mzuzu, Malawi for a total of 99 samples. Most of the latrines had a concrete floor or portable slab (85%, [28/33]) and the rest did not have a slab (15%, [5/33]). The mean number of people per household was 7.6 (standard deviation = 3.4, range = 3–18) and 74% (24/33) of households reported sharing the latrine with other households.

We commonly detected bacteria, viruses, and protozoa at each of the three pit latrine depths (Table 1), though we did not detect *Vibrio cholerae* or *Entamoeba histolytica*. Among the helminths, we did not detect *Trichuris trichiura* from any sample, but detected *Ascaris lumbricoides* in most samples (Table 1, Table 2).

### Rank order detection between latrines

At each depth, we observed the same rank order prevalence for the five most common bacterial pathogens: Enteroaggregative *E. coli* (EAEC), Enterotoxigenic *E. coli* (ETEC), *Shigella*/Enteroinvasive *E. coli* (EIEC), Enteropathogenic *E. coli* (EPEC), and *Campylobacter coli/jejuni*. Likewise, we detected all five viral pathogens in the same rank order at all three depths, except for sapovirus and adenovirus 40/41 in the surface samples, and we observed the same rank order detection among all three protozoan pathogens and both helminths (Table 2).

**Table 1. Detection of pathogens in fecal sludges (n = 33 pits).**

|  | Median number of pathogens detected per latrine (25th, 75th percentile)) | | | Prevalence of any pathogen (95% CI) | | |
|---|---|---|---|---|---|---|
|  | Surface (n = 33) | Mid-point (n = 33) | Maximum depth (n = 33) | Surface (n = 33) | Mid-point (n = 33) | Maximum depth (n = 33) |
| **Bacteria (out of 10)** | 3 (3, 4) | 3 (2, 4) | 3 (2, 4) | 100% | 100% | 100% |
| **Viruses (out of 5)** | 3 (2, 4) | 4 (3, 4) | 4 (3, 5) | 100% | 100% | 100% |
| **Protozoa (out of 3)** | 1 (1, 1) | 1 (1, 1) | 1 (1, 1) | 88% (77%, 99%) | 94% (86%, 100%) | 94% (86%, 100%) |
| **Helminths (out of 2)** | 1 (0, 1) | 1 (1, 1) | 1 (0, 1) | 55% (37%, 72%) | 76% (61%, 91%) | 67% (50%, 83%) |
| **All pathogens (out of 20)** | 8 (7, 9) | 9 (7, 11) | 8[a] (7, 10) | 100% | 100% | 100% |

Note: Pathogens detected via RT-qPCR using a custom TaqMan array card. 33 samples were analyzed at each pit latrine depth for a total of 99 samples.

(a) This is the median value, not the mean, so the value reported does not need to equal the sum of the individual taxa reported above.

**Table 2. Pathogen and indicator detection by pit latrine depth.**

| Pathogens | Pathogen detection | | | Jaccard similarity coefficient* | | |
|---|---|---|---|---|---|---|
| | Maximum depth (95% CI) | Mid-point (95% CI) | Surface (95% CI) | Surface- Mid-point | Mid-point- Maximum depth | Surface- Maximum depth |
| **Bacteria** | | | | | | |
| EAEC | 91% (81%, 100%) | 100% | 100% | 97% (91%, 100%) | 94% (85%, 100%) | 91% (79%, 100%) |
| ETEC | 70% (54%, 86%) | 76% (61%, 91%) | 79% (65%, 93%) | 82% (68%, 96%) | 78% (62%, 93%) | 75% (58%, 90%) |
| *Shigella*/EIEC | 70% (54%, 86%) | 64% (47%, 80%) | 67% (50%, 83%) | 65% (48%, 85%) | 76% (58%, 92%) | 67% (47%, 83%) |
| EPEC | 27% (12%, 43%) | 33% (17%, 50%) | 33% (17%, 50%) | 29% (5.9%, 50%) | 67% (38%, 92%) | 25% (8.3%, 53%) |
| *Campylobacter coli/ jejuni* | 18% (4.8%, 32%) | 27% (12%, 43%) | 27% (12%, 43%) | 29% (9.1%, 67%) | 50% (18%, 82%) | 36% (7.1%, 55%) |
| STEC | 12% (0.81%, 23%) | 6.1% (0%, 14%) | 9.1% (0%, 19%) | 25% (0%, 67%) | 25% (0%, 100%) | 20% (0%, 100%) |
| *C. difficile* | 9.1% (0%, 19%) | 6.1% (0%, 14%) | 6.1% (0%, 14%) | 100% | 67% (0%, 100%) | 67% (0%, 100%) |
| *Yersinia* spp. | 6.1% (0%, 14%) | 9.1% (0%, 19%) | 6.1% (0%, 14%) | 25% (0%, 100%) | 67% (0%, 100%) | 33% (0%, 100%) |
| *Salmonella* spp. | 3.0% (0%, 9.0%) | 9.1% (0%, 19%) | 9.1% (0%, 19%) | 20% (0%, 100%) | 33% (0%, 100%) | 33% (0%, 67%) |
| *Vibrio cholerae* | 0% | 0% | 0% | | | |
| **Viruses** | | | | | | |
| Norovirus GI/GII | 94% (86%, 100%) | 97% (91%, 100%) | 91% (81%, 100%) | 88% (73%, 97%) | 97% (90%, 100%) | 85% (76%, 97%) |
| Astrovirus | 88% (77%, 99%) | 88% (77%, 99%) | 82% (68%, 95%) | 81% (59%, 90%) | 93% (83%, 100%) | 75% (66%, 94%) |
| Sapovirus I/II/IV/V | 67% (50%, 83%) | 67% (50%, 83%) | 58% (40%, 75%) | 64% (50%, 86%) | 67% (48%, 84%) | 68% (44%, 83%) |
| Adenovirus 40/41 | 67% (50%, 83%) | 61% (44%, 78%) | 64% (47%, 80%) | 52% (46%, 83%) | 68% (48%, 86%) | 65% (33%, 71%) |
| Rotavirus A | 48% (31%, 65%) | 48% (31%, 65%) | 36% (20%, 53%) | 55% (32%, 79%) | 79% (57%, 91%) | 63% (25%, 70%) |
| **Protozoa** | | | | | | |
| *Giardia duodenalis* | 94% (86%, 100%) | 94% (86%, 100%) | 88% (77%, 99%) | 88% (70%, 94%) | 94% (84%, 100%) | 82% (75%, 97%) |
| *Cryptosporidium parvum* | 24% (9.4%, 39%) | 9.1% (0%, 19%) | 12% (0.81%, 23%) | 17% (0%, 50%) | 10% (0%, 33%) | 20% (0%, 50%) |
| *Entamoeba histolytica* | 0% | 0% | 0% | ND | ND | ND |
| **STHs** | | | | | | |
| *Ascaris lumbricoides* | 67% (50%, 83%) | 76% (61%, 91%) | 55% (37%, 72%) | 59% (35%, 73%) | 74% (57%, 89%) | 54% (41%, 78%) |
| *Trichuris trichiuria* | 0% | 0% | 0% | ND | ND | ND |
| | | | | | | |
| Overall | | | | 67% (62%, 72%) | 78% (73%, 82%) | 67% (61%, 72%) |

Note: ND: non-detect. *C. difficile*: *Clostridium difficile*. EAEC: Enteroaggregative *E. coli*. EIEC: Enteroinvasive *E. coli*. ETEC: Enterotoxigenic *E. coli*. EPEC: Enteropathogenic *E. coli*. STEC: shiga-toxin producing *E. coli*.

* Size of the intersection of matched detections divided by the size of the union of detections. For example, if we detected *Giardia duodenalis* in both the surface and the mid-point fecal sludge samples 28 times, only in the surface 1 time, only in the mid-point 3 times, and did not detect *Giardia duodenalis* in either sample 1 time: $J_{Giardia\ duodenalis}$ = (28) / (28 + 1 + 3) = 88%. 95% confidence intervals calculated by bootstrapping with 10,000 iterations.

## Comparison between sampling depths

Comparing the three sampling depths, the Jaccard similarity coefficient indicated that the maximum depth and mid-point samples were most similar ($J_{maximum,mid-point}$ = 78%,; 95% CI = 73%, 82%), followed by the surface and mid-point samples ($J_{surface,mid-point}$ = 67%; 95% CI = 62%, 72%), and surface and maximum depth samples ($J_{surface,maximum}$ = 67%; 95% CI = 61%, 72%). Co-detection of a pathogen at the three depths from the same pit latrine tended to increase with increased prevalence in sludges (S3 Table).

Adjusted for household population, latrine sharing, the presence of a concrete floor or slab, water source, and anal cleansing materials, there was no significant effect of depth on the prevalence of a pathogen on TAC ($X^2$(df = 2) = 2.4, p = 0.30). The odds of detecting the 20

**Table 3. Odds of detecting the 20 pathogens on TAC in samples from different pit depths.**

| Variable | Reference | OR (95% CI) | aOR (95% CI) |
|---|---|---|---|
| Depth: mid-point | Maximum depth | 1.1 (0.73, 1.6) | 1.1 (0.73, 1.6) |
| Depth: surface | | 0.81 (0.54, 1.2) | 0.80 (0.54, 1.2) |
| Household population | 1-person increase | **1.2 (1.1, 1.3)** | **1.2 (1.1, 1.3)** |
| Shared status: shared by multiple households | Private latrine | 1.8 (0.86, 3.9) | 1.3 (0.68, 2.3) |
| Water source: inside the compound | Communal public tap | 0.79 (0.36, 1.7) | **0.46 (0.23, 0.93)** |
| Water source: inside the house | | **0.28 (0.10, 0.76)** | **0.13 (0.05, 0.36)** |
| Anal cleansing: Water | Toilet paper | 0.76 (0.29, 2.0) | 1.8 (0.79, 4.2) |
| Concrete floor or slab present | No slab | 0.63 (0.24, 1.7) | 0.89 (0.42, 1.9) |

Note: OR: odds ratio. aOR: adjusted odds ratio. Bold indicates p ≤ 0.05

pathogens on TAC at the mid-point compared to the maximum depth was nearly one (aOR = 1.1; 95% confidence interval = 0.73, 1.6) and also nearly one for the surface compared to the maximum depth (aOR = 0.80, 95% CI = 0.54, 1.2) (Table 3). This result indicates that we did not observe a significant difference in detected pathogens between the three depths. Further, a 1-person increase in household population was associated with increased odds of pathogen detection (aOR = 1.2, 95% CI = 1.1, 1.3); latrines shared among multiple households were not associated with pathogen detection compared to private latrines (aOR = 1.3, 95% CI: 0.68, 2.3); both having a water source inside the compound (aOR = 0.46, 95% CI: 0.23, 0.93) and inside the home (aOR = 0.13, 95% CI = 0.05, 0.36) were associated with reduced odds of pathogen detection compared to households who reported a public tap as their water source; using water for anal cleansing was not associated with pathogen detection compared to using toilet paper (aOR = 1.8, 95% CI: 0.79, 4.2); and latrines with concrete floors or slabs were not associated with pathogen detection (aOR = 0.89, 95% CI: 0.42, 1.9) compared to latrines without concrete floors or slabs (Table 3).

## Discussion

We observed a high degree of homogeneity in enteric pathogen detection between fecal sludge samples collected at the surface, mid-point, and maximum depth of pit latrines in urban and peri-urban Malawi. Rank order prevalence, regression analysis and the Jaccard similarity coefficient indicated the pathogens detected in sludges from the maximum depth of latrines were not substantially different compared to pathogens detected in sludges from the mid-point and surface of the same pits. This finding suggests that a single fecal sludge sample taken from a standardized depth may be representative for enteric targets in a given pit. If near real-time pathogen surveillance is a goal, sampling the most recently deposited material near or at the surface of pit contents in a dry pit latrine would be justified and has the advantage of being the easiest strategy for sampling.

Though we lacked matched stools for comparison, the enteric pathogens we identified are known to be endemic in the study setting based on etiological studies of diarrheal disease [43–46]. For example, we frequently detected EAEC [45], norovirus [43,44,46], *Giardia duodenalis* [43,46], and *Ascaris lumbricoides* [47] in sludges and previous studies in Malawi also commonly detected these pathogens in children's stools. In addition, the individual pathogens we did not identify in fecal sludges aligned well with a study of 527 asymptomatic children in Blantyre, Malawi from 2013 to 2016 [45]. We did not detect *Vibrio cholerae*, *Trichuris trichiura*, or *Entamoeba hystolytica* in sludges, which Iturriza-Gómara *et al.* 2019 detected in 0%,

0.19% and 1.5% of children's stools, respectively. Given our relatively small sample size, detection of these three pathogens in fecal sludges was not expected.

The frequent co-detection of pathogens in samples from the same pit may be a result of several plausible factors. First, mechanistic forces (e.g. gravity, the addition of greywater, or insect movements) may result in the homogenization of fecal sludges. We most frequently co-detected pathogens from the maximum depth and mid-point samples, which would have been subject to gravity-induced mixing for a longer period than sludges at the surface. Second, as is common in other low-income urban communities [22], our results suggest a high burden of enteric infection in urban and peri-urban Malawi. Repeated [44,48,49] or persistent enteric infections [50,51] may be present; certainly pathogen targets we identified were being shed by many of those contributing waste to these onsite systems. Finally, pumping sludge through the Gulper likely helped to homogenize the mid-point and maximum depth samples [26] and centrifuging sludges may have further helped to concentrate pathogens.

We observed that increasing household population was associated with an increased odds of pathogen detection. This finding was expected as the more people contribute waste to a pit latrine the greater the probability that at least one person from the household would shed a specific pathogen into the pit. This may not be a generalizable observation, however: a previous study from a similar setting in Mozambique found no association between pathogen detection and number of individuals within housing compounds [24]. In addition, we found that having a water source inside the compound or home reduced the odds of pathogen detection compared to relying on water from a public tap. Having a water source inside the home may result in increased water use for personal hygiene or consumption, which a 2015 systematic review found was generally associated with reduced gastrointestinal infection and diarrheal disease [52].

Identifying common enteric pathogens in fecal sludges could also be informative about plausible exposures to these pathogens. The subsurface transport of pathogens from fecal sludges in latrines to groundwater has been demonstrated under certain hydrological and soil conditions [20]. Pit latrine users may be exposed to fecal sludges from overflowing pit latrines or a lack of facility maintenance, and pit emptiers may be exposed to fecal sludges during pit emptying [39,53]. Future quantitative estimates of pathogen concentrations in fecal sludges would be useful to estimate infection risks from these exposures [54].

There are several important limitations to this study. First, by removing the solid waste from some pits we may have inadvertently mixed the fecal sludge and we did not sterilize the Gulper between collection of the maximum depth and mid-point samples. Mixing before sample collection and cross contamination during sampling may have occurred and might explain why we observed that these two samples were more closely aligned with each other than with the surface samples. However, solid waste removal was necessary to use the Gulper, sterilization of the Gulper was not logistically practical, and the mass of fecal sludge flowing through the Gulper was substantially greater than the mass of fecal sludge that may have stuck to the walls of the Gulper tube. Next, we used a quantal assay that determined the presence or absence of each pathogen. There may be important differences in pathogen concentrations between the depths assessed, which could influence sampling methods for waste-based epidemiology efforts from pit latrines. Although we frequently co-detected pathogens, we cannot definitively explain the causal mechanism behind our findings. A next step would be to conduct tracer studies in latrines and septic tanks in LMICs to assess mixing inside these systems. Our sample processing methods used centrifugation and discarded the resulting supernatant. While we frequently detected all viral pathogens, it is plausible that these methods underestimated viral pathogen prevalence by not analyzing the supernatant as well as the solids. Further, numerous studies have characterized pathogen decay in wastewater, including the decay of

nucleic acids which are detected via PCR-based methods like those used in this study. Similar decay experiments are needed for fecal sludges. Additionally, we did not include household wealth or income as a confounder in our regression analysis because such an assessment would have added to the length and complexity of the questionnaire. It is likely some of the significant associations we found, such as the association between pathogen detection and water source, were confounded by socioeconomic status. Finally, as a cross sectional study, we were unable to assess pathogen signals over time, which are necessary for interpretation and use in surveillance efforts.

Though we most frequently co-detected pathogens in the matched samples from the mid-point and maximum depth of pits, sludge samples from the surface of pit latrines may be desired for surveillance efforts. Operating a Gulper is messy and requires specialized training. Other sampling methods exist to collect sludge from the mid-point or maximum depth of pit latrines [24,34], but still require specialized equipment (e.g. a Wheaton sub-surface sampler). The simplicity of using a hand scoop, or scoop like device such as a Sludge Nabber [24], to collect sludges from the surface of pit latrines offers an inexpensive and scalable WBE collection method. The sludge at or near the surface of pit latrine solids also represents the most recent waste contribution to the pit, which may be desired if a goal of the survey is to detect pathogens currently being shed by latrine users. Combined with customizable, multiplex molecular pathogen detection [55–57,58], sampling fecal sludges offers a compelling opportunity to gather novel and actionable public health data in LMICs where onsite sanitation predominates.

## Supporting information

**S1 Fig. Hand scoop for fecal sludge collection.**
(TIF)

**S2 Fig. Solid waste removal from a pit latrine.**
(TIF)

**S3 Fig. Operating the Gulper.**
(TIF)

**S4 Fig. Aliquoting a fecal sludge sample.**
(TIF)

**S1 Text. Methodology for total nucleic acid extraction from fecal sludges.**
(DOCX)

**S2 Text. Custom TaqMan Array Card (TAC).**
(DOCX)

**S1 Table. Assays used on the custom TAC.**
(DOCX)

**S2 Table. Interpretation of gene targets on the TAC.**
(DOCX)

**S3 Table. Co-detection by depth.**
(DOCX)

## Author Contributions

**Conceptualization:** Francis L. de los Reyes, III, Rochelle H. Holm, Elizabeth Tilley, Joe Brown.

**Data curation:** Drew Capone, Rochelle H. Holm.

**Formal analysis:** Drew Capone, Benjamin B. Risk.

**Funding acquisition:** Joe Brown.

**Investigation:** Petros Chigwechokha, Francis L. de los Reyes, III, Rochelle H. Holm, Elizabeth Tilley.

**Methodology:** Drew Capone, Petros Chigwechokha, Francis L. de los Reyes, III, Rochelle H. Holm, Benjamin B. Risk, Elizabeth Tilley, Joe Brown.

**Project administration:** Francis L. de los Reyes, III, Joe Brown.

**Supervision:** Francis L. de los Reyes, III, Elizabeth Tilley, Joe Brown.

**Writing – original draft:** Drew Capone.

**Writing – review & editing:** Petros Chigwechokha, Francis L. de los Reyes, III, Rochelle H. Holm, Benjamin B. Risk, Elizabeth Tilley, Joe Brown.

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
