## [Decision Letter · Decision Letter 0]

2 Jan 2021

Dear Dr. Brown,

Thank you very much for submitting your manuscript "Impact of sampling depth on enteric pathogen detection in pit latrines" for consideration at PLOS Neglected Tropical Diseases. As with all papers reviewed by the journal, your manuscript was reviewed by members of the editorial board and by several independent reviewers. The reviewers appreciated the attention to an important topic. Based on the reviews, we are likely to accept this manuscript for publication, providing that you modify the manuscript according to the review recommendations. 

Sincerely,

Poppy H L Lamberton

Deputy Editor

Poppy Lamberton

Deputy Editor

Reviewer's Responses to Questions

**Key Review Criteria Required for Acceptance?**

**Methods**

-Are the objectives of the study clearly articulated with a clear testable hypothesis stated?

-Is the study design appropriate to address the stated objectives?

-Is the population clearly described and appropriate for the hypothesis being tested?

-Is the sample size sufficient to ensure adequate power to address the hypothesis being tested?

-Were correct statistical analysis used to support conclusions?

-Are there concerns about ethical or regulatory requirements being met?

Reviewer #1: - Sample collection very well described. 

- Direct observation used to characterize latrine, good methodology to avoid bias. 

- Statistical anaylisis appropriate and well described. 

- Only a suggestion, instead of household population and whether the pit latrine was shared with other households, authors could adjusted by the number of members using the latrine (if available). 

- Main study limitations identified in discussion.

Reviewer #2: (No Response)

Reviewer #3: - The main objective was to evaluate the spatial distribution of enteric pathogens inside pit latrines with a clear and testable hypothesis.

-The study design is appropriate to address the stated objectives

-The population is clearly described and appropriate for the hypothesis being tested

-The sample size is sufficient to ensure adequate power to address the hypothesis being tested

-statistical analysis used to support conclusions are correct using rank order prevalence, regression analysis and the Jaccard similarity coefficient 

-There are no concerns regarding ethical or regulatory requirements.Ethical approval and material transfer agreement have been received from the Republicof Malawi National Health Sciences Research Committee

**Results**

-Does the analysis presented match the analysis plan?

-Are the results clearly and completely presented?

-Are the figures (Tables, Images) of sufficient quality for clarity?

Reviewer #1: - Table 1, In my opinion it is a bit confusing the total number of latrines sampled, the number of latrines tat were contaminated and the number of parasites detected. I would clarify that.

Reviewer #2: (No Response)

Reviewer #3: The analysis plan is matching perfectly with the final analysis.

The results are clear, complete and tables presented are sufficiet for clarity

**Conclusions**

-Are the conclusions supported by the data presented?

-Are the limitations of analysis clearly described?

-Do the authors discuss how these data can be helpful to advance our understanding of the topic under study?

-Is public health relevance addressed?

Reviewer #1: - Valuable interpretation of results. 

- Study limitations clearly stated.

Reviewer #2: (No Response)

Reviewer #3: Conclusions are in adequation with data presented

Authors described clearly the limitations of their analysis and explain our their founding will be helphul to implement Wastewater based epidemiology (WBE ) in pit latrines.

Public health relevance is adressed.

**Editorial and Data Presentation Modifications?**

Reviewer #1: (No Response)

Reviewer #2: (No Response)

Reviewer #3: (No Response)

**Summary and General Comments**

Reviewer #1: Relevant question. Problem and knowledge gap very well described and understandable.

Reviewer #2: Line 139 - 145: The SI units should be standardise across the document.

Line 148 - MS2 should be clearly defined

Line 196 - The total in column 3 doesn't seem to add up

Line 206 & 213 - I can't seem to understand the inter and intra latrine section. How are they defined?

Overall the study is ok. Future study should strongly factor eliminating cross-contamination. This study will compliment existing studies on pit latrines as it cut across regions using pit toilet systems.

Reviewer #3: This study provides interesting preliminary information on the collection methodology in small latrines. The methodological approach is good and the results are in line with the initial objective. It would be interesting to delimit the field of application of the results found.

Socio-economic, environmental (oxygen availability,temperature, moisture content, storage time, pH, volatile fatty acid) and microbiological factors were not taken into account in the variables analyzed and could constitute slight biases.

PLOS authors have the option to publish the peer review history of their article (what does this mean?). If published, this will include your full peer review and any attached files.

Reviewer #1: No

Reviewer #2: Yes: Olayinka Osuolale

Reviewer #3: Yes: Abdou Kader Ndiaye
---

## [Editor Report · Decision Letter 1]

19 Jan 2021

Dear Dr. Brown,

Thank you very much for submitting your manuscript "Impact of sampling depth on enteric pathogen detection in pit latrines" for consideration at PLOS Neglected Tropical Diseases. As editor I am happy with the changes you have made, but would like to see three very minor extra changes to this resubmitted manuscript:

1) add a space on line 82 (fewer than wastewater (25,26).Because fecal sludges can be sampled from shared latrines, pit )

2) I am happy with your response to Reviewer 2 on their point 3 about the median values in the table, and not the means, and hence why they don't add up. However, as other readers may also make a similar misunderstanding, please clarify in the legend that the values are medians within each row. 

3) Reviewer 3 point 1, Whilst this is outside the scope of your study, in your discussion please state that whilst this study focuses on the potential of this method for community/area surveillance for specific pathogens, use of these methods could also provide information on risks from these latrines, and then mention the three risks that they have suggested.

Sincerely,

Poppy H L Lamberton

Deputy Editor

Poppy Lamberton

Deputy Editor

Three very minor changes to this resubmitted manuscript:

1) add a space on line 82 (fewer than wastewater (25,26).Because fecal sludges can be sampled from shared latrines, pit )

2) I am happy with your response to Reviewer 2 on their point 3 about the median values in the table, and not the means, and hence why they don't add up. However, as other readers may also make a similar misunderstanding, please clarify in the legend that the values are medians within each row. 

3) Reviewer 3 point 1, Whilst this is outside the scope of your study, in your discussion please state that whilst this study focuses on the potential of this method for community/area surveillance for specific pathogens, use of these methods could also provide information on risks from these latrines, and then mention the three risks that they have suggested.
---

## [Editor Report · Decision Letter 2]

25 Jan 2021

Dear Dr. Brown,

We are pleased to inform you that your manuscript 'Impact of sampling depth on enteric pathogen detection in pit latrines' has been provisionally accepted for publication in PLOS Neglected Tropical Diseases.

Best regards,

Poppy H L Lamberton

Deputy Editor

Poppy Lamberton

Deputy Editor

---

## [Editor Report · Acceptance letter]

24 Feb 2021

Dear Dr. Brown,

We are delighted to inform you that your manuscript, "Impact of sampling depth on enteric pathogen detection in pit latrines," has been formally accepted for publication in PLOS Neglected Tropical Diseases.

Best regards,

Shaden Kamhawi

co-Editor-in-Chief

Paul Brindley

co-Editor-in-Chief
